# Large scale meta-analysis of preclinical toxicity data for target characterisation and hypotheses generation

Jordi Munoz-Muriedas *

Computational Toxicology, Data and Computational Sciences, GlaxoSmithKline, London, United Kingdom

* jordi.4.munoz-muriedas@gsk.com

## Abstract

Recent technological advances in the field of big data have increased our capabilities to query large databases and combine information from different domains and disciplines. In the area of preclinical studies, initiatives like SEND (Standard for Exchange of Nonclinical Data) will also contribute to collect and present nonclinical data in a consistent manner and increase analytical possibilities. With facilitated access to preclinical data and improvements in analytical algorithms there will surely be an expectation for organisations to ensure all the historical data available to them is leveraged to build new hypotheses. These kinds of analyses may soon become as important as the animal studies themselves, in addition to being critical components to achieve objectives aligned with 3Rs. This article proposes the application of meta-analyses at large scale in corporate databases as a tool to exploit data from both preclinical studies and in vitro pharmacological activity assays to identify associations between targets and tissues that can be used as seeds for the development of causal hypotheses to characterise of targets. A total of 833 in-house preclinical toxicity studies relating to 416 compounds reported to be active (pXC50 $\geq$ 5.5) against a panel of 96 selected targets of interest for potential off-target non desired effects were meta-analysed, aggregating the data in tissue–target pairs. The primary outcome was the odds ratio (OR) of the number of animals with observed events (any morphology, any severity) in treated and control groups in the tissue analysed. This led to a total of 2139 meta-analyses producing a total of 364 statistically significant associations (random effects model), 121 after adjusting by multiple comparison bias. The results show the utility of the proposed approach to leverage historical corporate data and may offer a vehicle for researchers to share, aggregate and analyse their preclinical toxicological data in precompetitive environments.

## Introduction

A critical part of the drug development process is the assessment of toxicity of a candidate at preclinical level in order to predict its safety profile. This is an area with high dependency on animal testing and still, despite the experimental effort, safety is a primary cause of attrition in the pharmaceutical industry. It is calculated that for each marketed drug, 50 research projects

**Data Availability Statement:** All relevant data are within the paper and its Supporting Information files. A subset of the GSK preclinical database involving the results covered with more detail in the paper (results of preclinical studies involving

Androgen Receptor agonists) is made available as supporting information. This dataset includes the results of 38 preclinical studies, annotated by species, administration route and duration, with the counts of animals with events observed in control and treated groups by tissue analysed. The data provided allows to reproduce the meta-analyses for the relations involving androgen receptors and provides an idea of the utility of the methodology the paper advocates for. The meta-analyses can be reproduced with the R code also provided.

**Funding:** Jordi Munoz-Muriedas conceived and designed the analysis, retrieved the data, performed the calculations, prepared conclusions and wrote the manuscript. The funder provided support in the form of salaries for Jordi Munoz-Muriedas but did not have any additional role in the study design, data collection and analysis, decision to publish, or preparation of the manuscript.

**Competing interests:** Jordi Munoz-Muriedas is a full-time employee of GlaxoSmithKline. Jordi Munoz-Muriedas has no other competing interests in relation to other companies, organisations or persons. GlaxoSmithKline is a global healthcare company with a portfolio of medicines in respiratory, HIV, immune-inflammatory and oncology therapeutic areas in addition to vaccines and healthcare products. Jordi Munoz-Muriedas confirms his commercial affiliation does not alter his adherence to all PLOS ONE policies on sharing data and materials.

will fail, with the main cause of attrition being non clinical toxicity in preclinical stages, safety in Phase I trials and efficacy, followed by safety, in Phase II trials [1].

Compared to other earlier stages in the drug development process, prediction of nonclinical toxicity or clinical safety is a big challenge for several reasons:

1. Nature of the data: As a drug discovery process progresses, the data evolves from being mainly quantitative in nature, homogenous and highly structured (for example, early stages are characterised by large production of quantitative numerical data like potency, physico-chemical properties and other, mainly in-vitro, assays that come often from a common protocol and are normally stored in centralised relational databases) to a more qualitative, heterogeneous and less structured data mainly stored in reports that may be scattered through different systems and formats (for example, narrative of a toxicity study, histopathological observations, images of lesions in tissues). Access and aggregation of such data for modelling becomes more difficult. Also, different ontologies may have been used to codify information at different stages, making aggregation even more difficult.

2. Availability of the data: As the drug discovery process progresses the assays and studies also become more complex and expensive and tend to be only performed on the most promising candidates, the results obtained becoming more strategic as opposed to transactional. This means that datasets coming from early stages often have a greater number of chemical structures with few information for each structure while datasets from later stages tend to have fewer structures with much information on each. This, together with the fact that the decision to obtain more strategic data is based on results of previous steps, means that a selection bias may be introduced.

3. Mechanistic understanding: Most of the areas in machine learning with recent advances are mainly based on statistical learning and are thus limited to the provision of signals based on tests for significance. However, statistical significance is not the same as biological plausibility.

In recent years, there have been technological advances that will help to overcome issues related with the nature of data, as for example, NoSQL (Not only SQL) databases that facilitate the management of diverse kinds of data and scale efficiently a amounts of information increase [2, 3], Natural Language Processing (NLP) methods, with potential to be used to quantitatively analyse text data [4, 5] and Convolutional Neural Networks in machine learning with potential to learn from histopathology images to make predictions [6, 7]. Regarding data availability, there has been an increase in the number of precompetitive collaborations among pharmaceutical companies in this space with the objective to share data and exploit it, in many cases in partnership with academic institutions and with incentive from governments, such as TOXCAST and eTOX [8]. These collaborations with exchange of data are also showing the urgent need for standardised controlled terminology and ontologies in preclinical toxicology. The introduction of the Standard for Exchange of Nonclinical Data (SEND), requiring companies to send in electronic format raw data from preclinical toxicology studies since December 2016, is expected to help in increasing both the volume of data available and its normalisation [9]. Finally, the introduction of frameworks such as the Adverse Outcome Pathways (AOP) is providing tools for practitioners to think and organise data in a causality manner [10, 11].

Despite progress made, one of the major problems that still confronts anybody trying to analyse and model preclinical data to build hypotheses is how to aggregate results from different animal studies. Animal studies are characterised by use of animals with low heterogeneity, but also by very small sample sizes due to cost, time, ethics and practicalities of the studies, which means those studies are generally underpowered or of unknown power [12]. While

meta-analysis is the gold standard to aggregate clinical studies and increase statistical power in clinical scenarios, it is not widely used in the preclinical space, although in the last two decades different initiatives have emerged to promote its application there. For example, in 2004 CAMARADES was established to support groups involved in the systematic review and meta-analysis of data from animal studies [13, 14], originally focusing on stroke but later extending to other diseases. The SYRCLE group (Systematic Review Centre for Laboratory animal Experimentation) also actively promotes and trains individuals in the conduct of systematic reviews of preclinical studies [15]. In the more concrete space of preclinical toxicity, interest in evidence-based methods is growing due to their potential to improve transparency, objectivity, consistency and reproducibility, and to inform decisions [16]. Evidence based methods may also prevent unnecessary duplication of experiments, reduce animal testing by making further use of the data already available, refine animal tests by providing evidence to choose a particular animal model over another, suggest tests with lower burden of pain or shorten duration of the tests [17]. In addition, such analyses can be used to explore sources of heterogeneity, identify biases and generate hypotheses [18].

With the current revolution that the pharmaceutical industry is experiencing in the fields of data analytics and machine learning, it is possible that sooner rather than later data experiments to analyse and model the available preclinical data will be as important as, or even more important than, the conduct of new experiments based on animals in the area of preclinical research, with high expectations that our capability to predict the safety of drug candidates will improve. Moreover, there is an increasing expectation for researchers to use all former and ongoing research data available to conduct new research, as summarised by Lund et al. in "The evidence based research statement", hence, the urgent need for systems to aggregate and critically appraise the available information [19].

This article proposes a large-scale application of meta-analysis to combine corporate in vivo preclinical toxicity and in vitro pharmacology databases to mine for associations relating targets with events observed in in vivo studies. This approach may be useful to characterise the safety profile of targets and also to generate statistical hypotheses that can be used as seeds to direct efforts in the building of AOPs.

## Methods

### Information sources and data collection

A list of 97 targets of interest for safety purposes (referred in this article "the list" and available in S1 File) was compiled from two literature references [20, 21].

Histopathology animal data was extracted from historical preclinical toxicity studies available in GSK in-house database with records organised at observation level (all studies in the database were conducted in accordance with the GSK Policy on the Care, Welfare and Treatment of Laboratory Animals and were reviewed the Institutional Animal Care and Use Committee either at GSK or by the ethical review process at the institution where the work was performed). For each observation, among other details, the record includes information such as the nature of the observation (e.g. necrosis, infiltrate), its location (e.g. Tissue), its severity on a scale of 1 to 5, reference of the animal where it was observed, its species and strain, references of the dosing group and study the animal belongs to, name of compound tested, dosing regime, administration route, duration of the study and date and site of the experiments.

Pharmacology data against targets in "the list" for compounds for which preclinical data was available (identified in the previous step) was extracted from our in-house database of in vitro tests. Only data from functional assays was used, and activity concentrations 50 (inhibition or effect) were transformed into logarithmic scale (pIC50 or pEC50). Averages of

logarithmic values were taken if a compound was tested in more than one functional assay for the same target.

Data extraction and curation was performed by means of Standard Query Language (SQL) queries embedded in in-house Python code.

### Inclusion criteria and variables

The studies considered for meta-analyses were those (of any duration or administration route and involving any species) available for compounds with an in vitro measured pEC50 or pIC50 equal or greater than 5.5 for the target and mode of action analysed. Studies with combination of compounds or without control group were excluded.

For each study, the number of events in treated and untreated groups were counted per tissue considered, an event being defined as an animal with at least one visible lesion of any morphology and of any severity.

### Synthesis of results

Meta-analysis for each target and mode of action in "the list" was performed against all tissues explored in the preclinical studies included in the analysis. All tissues names were mapped to a controlled vocabulary. In order to facilitate visualisation of results, tissues were group following the Standard Organs Classification (SOC) where possible.

Aggregated odds ratios (OR) to assess the difference of prevalence of events in treated and untreated animals were obtained, along with their 95% confidence intervals and p values, applying both a Mantel-Haenszel fixed effect model and a Paule-Mandel random effects model with Q-profile for calculation of confidence interval for inter-studies variance [22–24]. Studies with zero events were included and the Haldane-Anscombe correction applied for empty cells. A meta-analysis would only be performed if there were data coming from at least three different compounds and three studies per tissue/(target-mode of action) combination analysed.

Additional meta-analyses of subgroups of studies aggregated by administration route, species and duration of study were carried out in order to assess the effect of these covariates. In the case of administration route, the routes considered were oral ("PO"), intravenous ("IV"), dermal ("DERMAL") and inhaled ("RESPIR") with the remaining routes grouped under "OTHER" category. In the case of duration, studies were classified as short (recorded duration less than 9 days) and long (more than 9 days).

Heterogeneity was explored by means of a chi-squared test. Given the problem of the test for heterogeneity being too specific for small datasets and overpowered for large datasets, the $I^2$ statistical is also calculated to assess the impact of the heterogenicity among studies (heterogeneity was considered high if $I^2$ greater than 0.75, moderate if $I^2$ between 0.5 and 0.75 and low between 0.25 and 0.5) [25].

Meta-analysis calculations were performed in R using the "meta" package version 4.11.0 [26]. All forest plots were created with the same meta package except plot in Fig 4, which was generated using the "forestplot" package version 1.9 [27]. Version of R used was 3.6.2, "Dark and Stormy Night" [28].

### Control for multiple comparison bias

Multiple comparison bias in the generation of hypothesis for AOPs was controlled by adjusting the p-values obtained in the Meta-analyses using the Benjamini-Hockberg method as implemented in R ("p.adjust" function), with a 5% threshold for the False Discovery Rate (FDR) [29].

## Results

### Descriptive analysis

The following flow diagram provides a summary of the process followed (Fig 1).

The data queries returned a total of 831 studies available for analysis, involving 414 compounds that were active for at least one target in "the list".

These studies represented a total of 24,358 animals (17,256 treated and 7,102 controls) and a total of 232,352 histological observations across 271 tissues. Most of the studies have a small number of animals, with more than a half of them including less than 20 animals, often distributed in three dosing groups and one control group (more details about the distribution of animals per study in S1 Fig).

The preclinical data available is very sparse when it comes to tissue observations available for every study: out of a theoretical full matrix with 225,743 tissue-study pairs (271 tissues x 833 studies) only 15,613 (6.9%) pairs were available. A group of 8 tissues that was consistently observed in 648 of the 833 studies accounted for a 33% of all the data available, while a group of an additional 38 tissues consistently measured in 179 studies accounted for an additional 40% of the data available.

Most of the preclinical data comes from studies in rat and has an oral route of administration. In terms of duration, a large proportion of the studies (42%) had durations of less than 10 days (mainly 7 days toxicity studies by protocol) while another 45% of the data was formed by studies of durations between 11 and 40 days (mainly 14 days and 28 days studies by protocol). Fig 2 summarises the distribution of the preclinical data.

The pharmacological data obtained for the compounds involved in the studies is also very sparse. A total of 770 compounds with preclinical data available had been tested against at least one of 71 out of the 96 targets included in "the list". Out of the theoretical full matrix of 54670 measures (770 compounds x 71 targets), 19,522 (36%) compound-target pairs were available. Most of the compounds (671) had data against a set of 25 targets and those measurements accounted for 85% of the data. After applying the activity threshold (pXC50 $\geq$ 5.5), there were only 414 compounds with activity greater than the threshold for at least one of 56 targets, resulting in a total of 978 compound-target interactions (without accounting for mode of action).

The merging of the in vivo preclinical and in vitro data sources to relate targets to tissues resulted also in a very sparse dataset. Of the theoretical maximum of 19,241 target–tissue possible pairs (71 targets x 271 tissues) only 1896 pairs had enough information available to fulfil the criteria to be included in the meta-analysis, reducing the number of targets to 48 and the number of tissues to 67. More specifically, the same group of 8 tissues mentioned earlier had data available for a set of 46 targets (all the surviving targets but 2), accounting for 20% of the data, and an additional set of 40 tissues had data for 30 targets, accounting for another 62% of the data.

The 1896 target-tissue pairs increased to 2139 pairs when the mode of action (agonist or antagonist) was considered, i.e. when different modes of action for the same target were counted as different targets.

Preclinical studies available for each target-tissue pair were grouped together with the objective to aggregate results using a meta-analysis. In many cases, the number of studies available for a target-tissue pair was small, with approximately 40% of the target-tissue pairs having less than 10 studies available (S2 Fig).

### Meta-analyses

**Significant associations.**    A total of 2139 meta-analyses were carried out, producing 612 significant associations (i.e. target-tissue combinations for which there was a significant effect

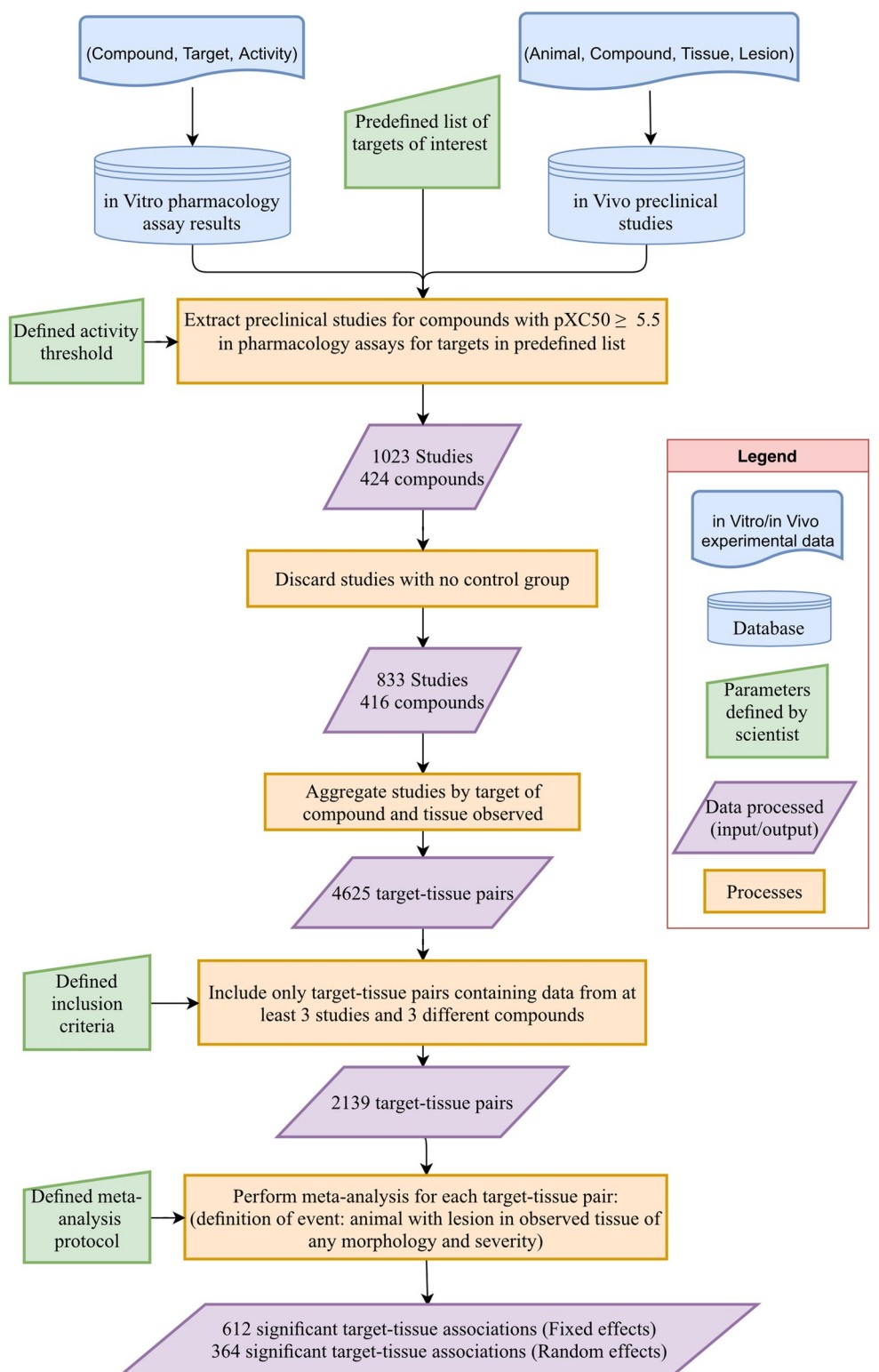

**Fig 1. Flow diagram with the steps in the analysis.**

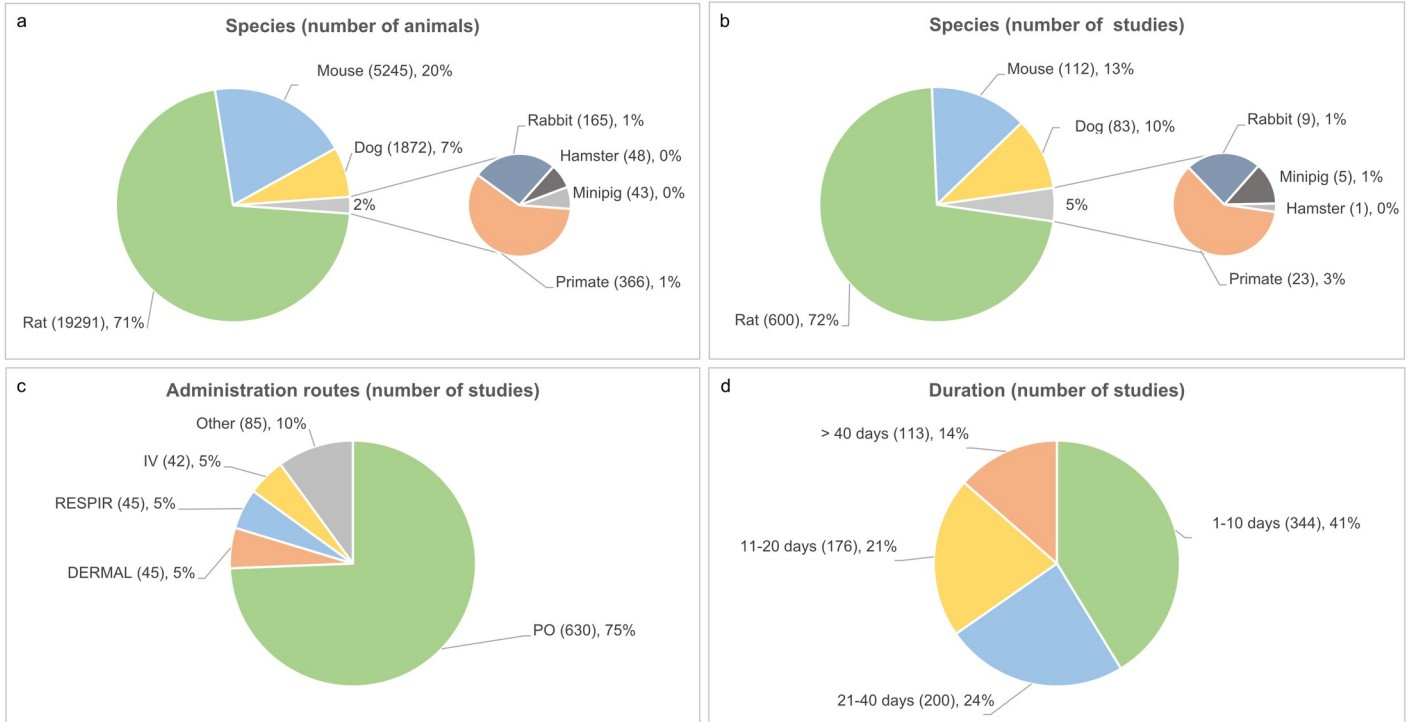

**Fig 2. Distribution of species, administration routes and duration across studies.**

of the compound on the number of events), involving 46 targets (63 taking into account mode of action) and 54 tissues, at p ≤ 0.05 using a fixed effect model, and 364 significant associations, involving also 46 targets (62 taking into account mode of action) and 42 tissues, when a random effects model was applied. Control of multiple comparison bias by applying the Benjamini-Hochberg adjustment with a false discovery rate of 5% reduced the number of significant associations to 304 in the case of the fixed effect model, involving 45 targets (58 taking into account mode of action) and 41 tissues, and to 121 in the case of the random effects model, involving 34 targets (41 taking into account mode of action) and 24 tissues. S3 and S4 Figs summarise the number of significant target-tissue associations obtained after applying different methods to control for multiple comparison bias. An Excel spreadsheet file is available in supporting information with the significant associations obtained with the random effects model (S2 File).

**Heterogeneity and subgroup differences.** Heterogeneity in the meta-analyses was generally low, with no heterogeneity observed in 80% of the meta-analyses performed, and only a 5% of the meta-analyses showing an $I^2$ larger than 50% (S5 Fig).

Subgroup meta-analyses taking account of possible differences in effect size between the species, administration routes and duration of the studies found a low number of cases where differences across group were significant. Significant differences across groups were found only in 5.5% of the meta-analyses between species, in 3% of the meta-analyses between administration routes, and in 3% of the meta-analyses between short and long duration. Low numbers are expected given that in many cases, no events or a low number of events are observed in the studies (almost half of the meta-analysis produced an OR of 1). If we confine attention to the meta-analyses with higher size of effect (OR ≥ 2), the number of significant differences across groups increases for species and administration route (15% of meta-analyses had

significant differences across species, 11% across administration routes) but remains the same in the case of duration (3%). S9 Fig summarises the number of differences identified across covariates.

**Fixed effect vs random effects.** Both random effects and fixed effect models showed in general good agreement on the size of the effect (S6 Fig), especially in those situations where heterogeneity was low, although a slight trend to produce larger OR was observed in the fixed effect models. In 50% of the cases, the increase in ln(OR) was less than 0.06 and in 75% of the cases it was less than 0.16. Fixed effect and random effects models agreed in giving the estimated OR of the effect to be $\geq 2$ in 407 target-tissue pairs (19% of the total number of meta-analysis) and $<2$ in 1582 (74% of the total). The main disagreement between models was for a set of 131 target-tissue pairs (6%) classified as OR $\geq 2$ by the fixed effect models but not by the random effects model. On the other hand, only 19 pairs (1% of the total) were classified to be OR $\geq 2$ by the random effect model but not by the fixed effect model (Cohen's kappa value for the agreement between random and fixed in classifying OR above or below 2 is 0.80, where $\kappa$ = 0 represents agreement no better than by chance and $\kappa$ = 1 represents complete agreement). The agreement is still high after excluding the 1948 meta-analyses with low heterogeneity ($I^2 < 30\%$), both fixed effect and random effects models giving estimated OR > 2 in 107 meta-analysis (56% of the remaining total number of meta-analyses) and to be less than 2 in 45 meta-analysis (24%) ($\kappa$ = 0.54).

The main difference between random and fixed effect models was in the confidence interval, which, as expected, are larger in the random effects models. Also as expected, the increase, is small in situations with low heterogeneity ($I^2 < 30\%$), where in 75% of the cases the increase in the confidence interval is less than 12%, whereas in situations with larger heterogeneity the increase is larger than 58% in half of the meta-analysis and more than 100% in 25% of the cases, reducing dramatically the number of results considered significant.

## Summary of key findings

Fig 3 summarises all the results for meta-analyses where the OR obtained for the target–tissue association was $\leq 2$. Tissues are distributed in columns and targets in rows and the data points are coloured by the magnitude of the p-value for the interaction, those in red being the results that would be considered significant after adjusting by Benjamini-Hochberg method and those in yellow, the results that did not achieve statistical significance on this criterion. This figure may be useful to spot relations among the targets and tissues, to identify clusters of results or patterns of activity. For example, looking column by column (tissues), the figure seems to indicate thymus, stomach and liver are some of the organs accumulating more significant events in preclinical animal studies. In the particular case of thymus, after applying the Benjamini-Hochberg adjustment, the results show association mainly with receptors related with neurotransmitters and hormones (AR, ADRA1B, ADRB2, CHRM1, CHRNA1, HTR1A, HTR2A, HTR3A, SLC6A3, SLC6A4) apart from a calcium channel (CACNA1) and a tyrosine-protein kinase (LCK). Also interesting is the cluster of associations highlighted between mammary gland and targets related with dopamine, serotonin, histamine and adrenaline in addition to the androgen receptor, although only DRD2 (Dopamine receptor 2) and AR (Androgen receptor) would be significant after adjusting the p-values. Looking row by row (targets), the figure can be used to characterise targets, for example, in the case of androgen receptors, the figure indicates events in the immune system, the adrenals, stomach, skin, femur, incisor and almost all the reproductive tissues observed, while in the case of PPARG (Peroxisome Proliferator Activated Receptor Gamma), the events concentrate in brown adipose tissue, stomach, liver, femur and sternum-bone marrow.

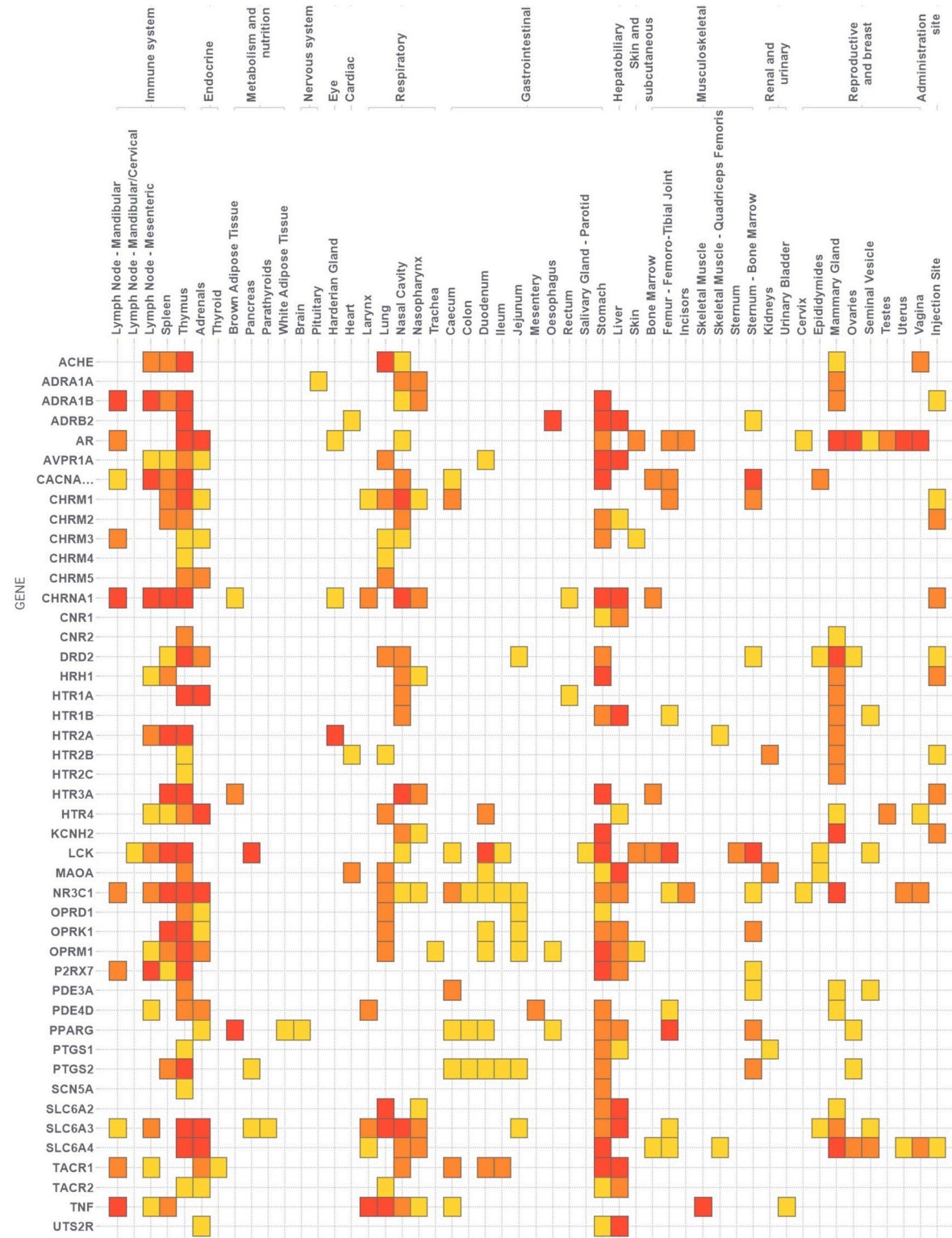

**Fig 3. Summary of largest effect sizes obtained (OR ≥ 2).** The plot contains all the target-tissue associations (regardless of mode of action) for which a meta-analysis produced an OR ≥ 2. Colours are based on p-values (yellow: not significant at p ≤ 0.05, orange: significant at p ≤ 0.05 but with FDR > 5%, red: significant) and with FDR ≤ 5%.

## Target profiles

Fig 3 only summarises the largest size effects but the results provided by the meta-analyses can be exploited to obtain more detailed analysis. The OR obtained in the meta-analyses and their confidence intervals can be grouped by target in a multiple forest plot to obtain what could be described as a 'target profile' over the range of tissues. Fig 4 shows an example of a target profile plot for agonists of the Androgen receptor. To facilitate visualisation, ORs are displayed as natural logarithms in order to keep confidence intervals symmetrical around the OR and facilitate the comparison of disparate values. In addition, related tissues are grouped by System Organ Classification (SOC). In the case of Androgen agonists, significant OR (before controlling for multiple comparison bias) are obtained for Thymus, Adrenals, Stomach, Incisors, and several reproductive tissues.

Meta-analysis can also be used to explore the consistencies in the results for a given target-tissue association across the different covariates in the studies, by means of subgroup meta-analysis. Those analyses can also be visualised by means of forest plots which offer a way to evaluate consistencies across groups that is quick and easy to interpret, helping to identify potential sources of heterogeneity and to identify studies with odd results. Fig 5 presents aggregate results grouped by species, administration routes, and durations, for the association between agonists of the androgen receptor and events in testes, which is one of the meta-analysis results with higher heterogeneity ($I^2 = 63\%$).

The subgroup meta-analysis at species level shows significant heterogeneity between studies on rat and on dog ($p < 0.01$ for subgroup differences). In fact, a significant effect of compounds on number of events, according to the random-effects model, is only observed within the studies on rat. Borderline heterogeneity ($p = 0.05$) was observed at administration route level, with the effect of compound on number of events coming mainly from oral studies (S5 Fig). Good agreement was observed when comparing short and long studies, with no significant heterogeneity observed ($p = 0.78$) (S6 Fig).

Additionally, a forest plot with all the studies facilitates the inspection of consistency across all the results obtained in the preclinical studies (S7 Fig).

The presence of bias can be assessed by means of a funnel plot such as the one in S8 Fig for the case of androgen receptor agonists–testes association. In this example, no marked biased can be inferred from the visualisation of the data points in the plot, and this is also indicated by the lack of significance in the Egger test ($p = 0.65$).

# Discussion

## Potential applications

The results of the meta-analysis presented here only represent statistical associations between targets and an increment in the number of events in the tissues explored, and this does not imply that the targets are responsible for the events as a statistical association does not necessarily imply causation. Moreover, the individual preclinical studies that are included in the meta-analyses were originally designed to show whether the compounds are responsible for the events. For that reason, it is important to apply a weight of evidence approach, examining the magnitude of the effect, the dose response and the biological plausibility of the events given the pharmacology of the compounds tested. A systematic review of literature available should be carried out to explore the biological plausibility of the association and gather the information needed to establish a causal explanation which could be represented in the form of an Adverse Outcome Pathway (AOP). In that sense, the results of the meta-analysis can be used as seeds to develop AOPs, suggesting potential links between molecular initiating events and

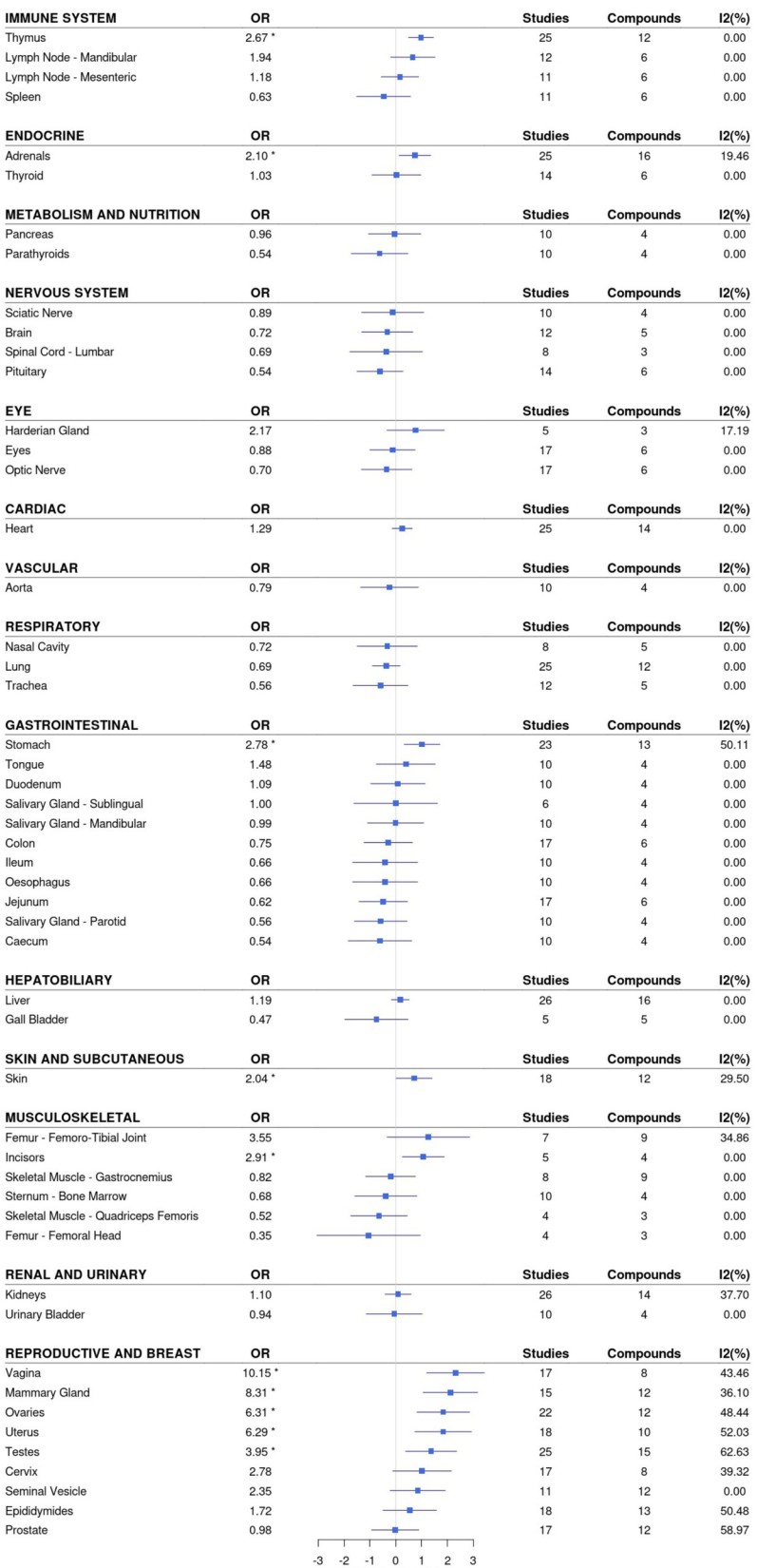

| IMMUNE SYSTEM | OR | | Studies | Compounds | I2(%) |
|---|---|---|---|---|---|
| Thymus | 2.67 * | | 25 | 12 | 0.00 |
| Lymph Node - Mandibular | 1.94 | | 12 | 6 | 0.00 |
| Lymph Node - Mesenteric | 1.18 | | 11 | 6 | 0.00 |
| Spleen | 0.63 | | 11 | 6 | 0.00 |
| **ENDOCRINE** | **OR** | | **Studies** | **Compounds** | **I2(%)** |
| Adrenals | 2.10 * | | 25 | 16 | 19.46 |
| Thyroid | 1.03 | | 14 | 6 | 0.00 |
| **METABOLISM AND NUTRITION** | **OR** | | **Studies** | **Compounds** | **I2(%)** |
| Pancreas | 0.96 | | 10 | 4 | 0.00 |
| Parathyroids | 0.54 | | 10 | 4 | 0.00 |
| **NERVOUS SYSTEM** | **OR** | | **Studies** | **Compounds** | **I2(%)** |
| Sciatic Nerve | 0.89 | | 10 | 4 | 0.00 |
| Brain | 0.72 | | 12 | 5 | 0.00 |
| Spinal Cord - Lumbar | 0.69 | | 8 | 3 | 0.00 |
| Pituitary | 0.54 | | 14 | 6 | 0.00 |
| **EYE** | **OR** | | **Studies** | **Compounds** | **I2(%)** |
| Harderian Gland | 2.17 | | 5 | 3 | 17.19 |
| Eyes | 0.88 | | 17 | 6 | 0.00 |
| Optic Nerve | 0.70 | | 17 | 6 | 0.00 |
| **CARDIAC** | **OR** | | **Studies** | **Compounds** | **I2(%)** |
| Heart | 1.29 | | 25 | 14 | 0.00 |
| **VASCULAR** | **OR** | | **Studies** | **Compounds** | **I2(%)** |
| Aorta | 0.79 | | 10 | 4 | 0.00 |
| **RESPIRATORY** | **OR** | | **Studies** | **Compounds** | **I2(%)** |
| Nasal Cavity | 0.72 | | 8 | 5 | 0.00 |
| Lung | 0.69 | | 25 | 12 | 0.00 |
| Trachea | 0.56 | | 12 | 5 | 0.00 |
| **GASTROINTESTINAL** | **OR** | | **Studies** | **Compounds** | **I2(%)** |
| Stomach | 2.78 * | | 23 | 13 | 50.11 |
| Tongue | 1.48 | | 10 | 4 | 0.00 |
| Duodenum | 1.09 | | 10 | 4 | 0.00 |
| Salivary Gland - Sublingual | 1.00 | | 6 | 4 | 0.00 |
| Salivary Gland - Mandibular | 0.99 | | 10 | 4 | 0.00 |
| Colon | 0.75 | | 17 | 6 | 0.00 |
| Ileum | 0.66 | | 10 | 4 | 0.00 |
| Oesophagus | 0.66 | | 10 | 4 | 0.00 |
| Jejunum | 0.62 | | 17 | 6 | 0.00 |
| Salivary Gland - Parotid | 0.56 | | 10 | 4 | 0.00 |
| Caecum | 0.54 | | 10 | 4 | 0.00 |
| **HEPATOBILIARY** | **OR** | | **Studies** | **Compounds** | **I2(%)** |
| Liver | 1.19 | | 26 | 16 | 0.00 |
| Gall Bladder | 0.47 | | 5 | 5 | 0.00 |
| **SKIN AND SUBCUTANEOUS** | **OR** | | **Studies** | **Compounds** | **I2(%)** |
| Skin | 2.04 * | | 18 | 12 | 29.50 |
| **MUSCULOSKELETAL** | **OR** | | **Studies** | **Compounds** | **I2(%)** |
| Femur - Femoro-Tibial Joint | 3.55 | | 7 | 9 | 34.86 |
| Incisors | 2.91 * | | 5 | 4 | 0.00 |
| Skeletal Muscle - Gastrocnemius | 0.82 | | 8 | 9 | 0.00 |
| Sternum - Bone Marrow | 0.68 | | 10 | 4 | 0.00 |
| Skeletal Muscle - Quadriceps Femoris | 0.52 | | 4 | 3 | 0.00 |
| Femur - Femoral Head | 0.35 | | 4 | 3 | 0.00 |
| **RENAL AND URINARY** | **OR** | | **Studies** | **Compounds** | **I2(%)** |
| Kidneys | 1.10 | | 26 | 14 | 37.70 |
| Urinary Bladder | 0.94 | | 10 | 4 | 0.00 |
| **REPRODUCTIVE AND BREAST** | **OR** | | **Studies** | **Compounds** | **I2(%)** |
| Vagina | 10.15 * | | 17 | 8 | 43.46 |
| Mammary Gland | 8.31 * | | 15 | 12 | 36.10 |
| Ovaries | 6.31 * | | 22 | 12 | 48.44 |
| Uterus | 6.29 * | | 18 | 10 | 52.03 |
| Testes | 3.95 * | | 25 | 15 | 62.63 |
| Cervix | 2.78 | | 17 | 8 | 39.32 |
| Seminal Vesicle | 2.35 | | 11 | 12 | 0.00 |
| Epididymides | 1.72 | | 18 | 13 | 50.48 |
| Prostate | 0.98 | | 17 | 12 | 58.97 |

-3  -2  -1  0  1  2  3
lnOR Random Effects

**Fig 4. Target profile forest plot for androgen receptor agonists (random effects).** The meta-analyses can be grouped by target to produce profiles of events per tissue. The figure shows the profile obtained for the meta-analyses involving androgen receptor agonists for each tissue analysed. The results of the meta-analyses allow to identify the tissues where there was a significant increase in number of animals with events vs controls.

effects (the two ends of the adverse outcome pathway). Meta-analysis results could also act as a heuristic function to prioritise research towards generation of AOPs (e.g. to guide literature reviews or exploration of knowledge graphs).

It is not the purpose of this article to establish such causal relations or generate AOPs, but to present a methodology to generate the associations. However, it is important to gather some insight on how likely this technique is to identify true associations between targets and tissues. Given that all the targets included in the analysis presented here are well characterised it is possible to assess if the effect on frequency of events of compounds known to hit those targets are in agreement with what would be expected from what is already known about the targets.

In the case of the associations related to thymus, it is interesting to find neurotransmitters at the top of the associations found by the random-effects models. It is important to take into account that the conditions of toxicity studies often induce stress in animals, with downstream effects on food consumption and activity of the animal, and will produce alterations in the immune and endocrine systems, especially on tissues like thymus, spleen and adrenals [30]. In the results presented, all studies included in the meta-analysis included control groups that underwent the same stress and the OR in each study should reflect the active compound vs. control comparison, potentially offsetting the component due to study design. While the meta-analyses did not have enough resolution to treat each dosing group as separate and establish a dose effect relationship, the size of the effects of the events that are considered significant after controlling by FDR is generally high, with OR above 2. From a biological perspective, it is recognised that the central nervous, the immune and the endocrine systems are interconnected through dense nerve and hormonal pathways [31]. Androgen receptors are found in the thymic epithelium and are postulated to modulate thymus size and thymocyte development and accelerate thymocyte apoptosis [32, 33]. B2 adrenergic receptors have been identified in the medulla of the rat thymus gland with experiments suggesting their relation with thymocyte proliferative response, histological changes and thymopoiesis [34–36]. Muscarinic receptors have also been reported in thymocytes and suggested to appear in late periods of cell maturation [37]. Additionally, the developing thymus of rat contains serotonin receptors and transporters and it has been suggested that those play a role on thymic development [38].

The associations of androgen receptors with reproductive tissues and adrenal glands was expected, as it was the association of brown adipose tissue with Peroxisome Proliferator Activated Receptor Gamma (PPARG), given its role in brown adipocyte differentiation required for the development and function of the tissue [39].

The relationship of dopamine with the mammary gland is expected, given that dopamine is the primary regulator, through its interaction with lactotroph D2 receptors, of the production in the pituitary gland of prolactin, which regulates the development of the mammary gland [40, 41]. In the case of a potential role for serotonin, that is not that well established but there are suggestions that the mammary gland contains a complex serotonergic regulatory system that plays an important role in the processes of mammary homeostasis and early involution [42]. Histamine is postulated to be involved in pregnancy-associated growth of the mammary gland in preclinical species [43]. Adrenergic α and β receptors can be found in the mammary gland, with reports that α adrenergic stimulation produces changes in milk yield and peak flow rate, indicating a function for adrenalin in the gland [44, 45].

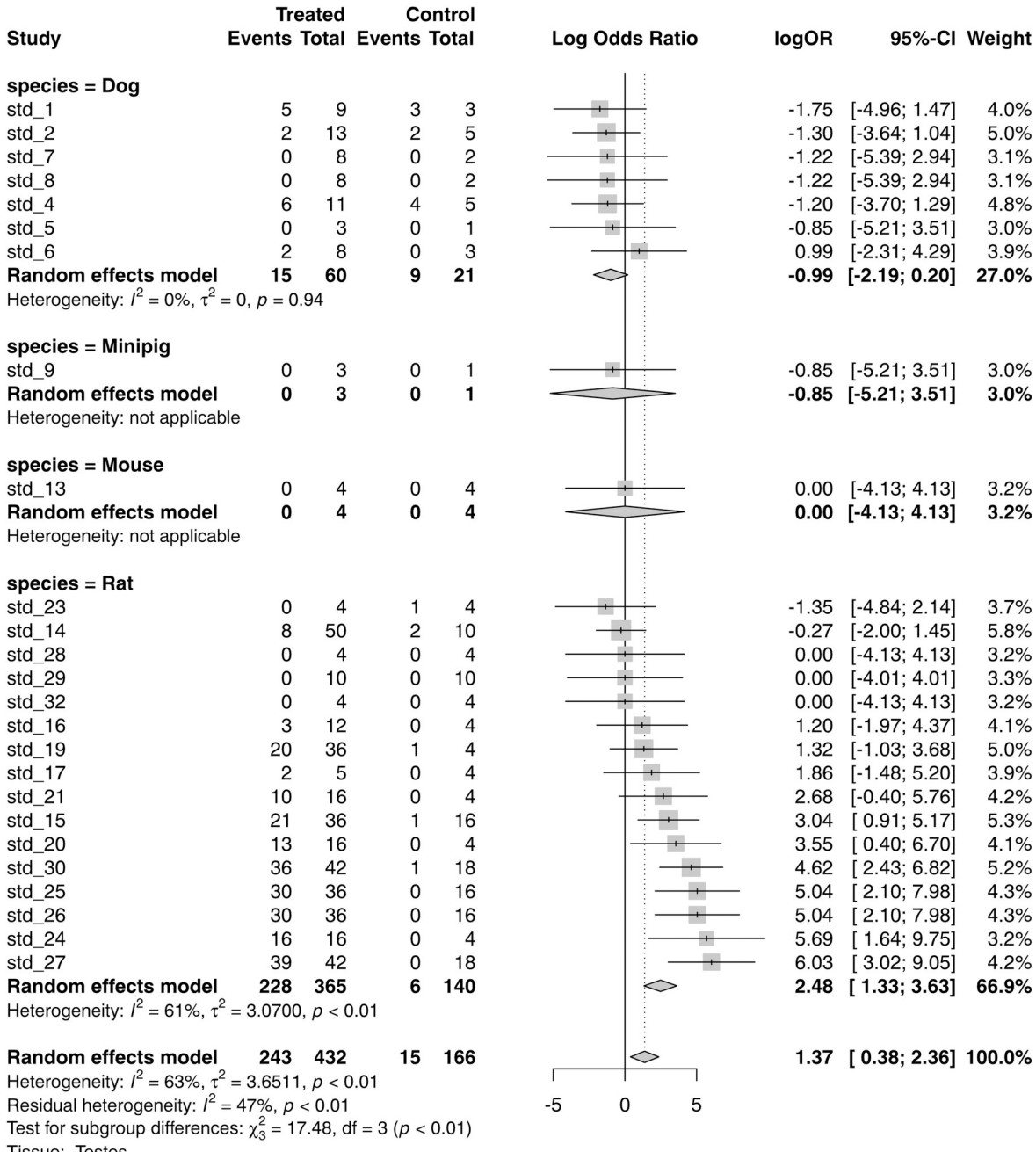

**Fig 5. Androgen agonists subgroup analysis for the effect on number of events in testes (random effects): Forest plots with results by species and tests for subgroup differences.** Forest plots can be used to evaluate the consistency of the effect across covariates and studies.

## Fixed effect vs random effects & heterogeneity

One of the decisions to make when applying a meta-analysis is whether to apply a fixed effect or a random effects model. Under the fixed effect paradigm, it is assumed that one true effect size is shared by all the combined studies, and variations across studies are due to sampling from a same distribution (i.e. if all studies had infinite sample size all of them would produce

the same result and error would be zero). Under the random effects paradigm, it is accepted that the true effect can vary from one study to another as samples may be coming from different distribution (i.e. even if studies had infinite sample sizes, i.e. they would produce different results and the error of the mean effect, averaged over studies, would not be zero). In other words, it is assumed that differences among observed outcomes are not only a result of random sampling fluctuations but also caused by random variability between studies, which is reflected by the presence of heterogeneity [46].

In general, low heterogeneity is observed in the meta-analyses presented in this publication: 90% of them had an $I^2$ lower than 30% and only a 4% of them had an $I^2$ greater than 50%. However, it is important to keep in mind that tests for heterogeneity may be underpowered, especially in those meta-analyses where the number of studies included is small and variance within study is large, a situation which is frequent in preclinical studies as they are usually underpowered due to its small sample size. Additionally, a non-significant p-value for heterogeneity is not proof that the studies share a common effect.

Animal preclinical studies are expected to have more homogenous populations than, for example, clinical trials in humans, due to the use of inbred strains and a better control of exposures, test conditions and outcome assessments [47]. However, heterogeneity in the meta-analysis can be introduced due to the combination of different species or by variation in study design and protocol (e.g. duration, administration route) although it may be justified to combine studies using different species if there is evidence that the outcome of interest works by the same mechanism across species or if differences are accounted for in the statistical model [48].

In any case, indications of low heterogeneity should not be used as justification for choosing a fixed effect model and the decision on what model to use should be based on whether there are any reasons to believe that the true effect is going to be the same in all studies. For that reason, and given that it may be difficult to assume that, for example, different species will share a common true effect or that the effect is going to be same regardless of duration of the treatment, it may be advisable to apply random effects models when all the studies are combined together and also explore the effect of co-variates like species, duration and administration by subgroup meta-analyses to assess consistency of results and explore potential sources of heterogeneity.

However, one problem may arise in preclinical scenarios when applying random effects models: if the number of studies in the meta-analysis is small, the estimation of between studies variance critical in random effects models will have poor precision. Nowadays, there seems to be a trend in literature suggesting as default approaches to random effects models the Paule-Mandel and the REML methods, especially now that they are available in many software packages [49–51]. With the REML method, convergence to a solution is not always guaranteed. On the other hand, the Paule Mandel method doesn't require convergence, and has additional advantages as no presuming a particular distribution for the data, and was selected as the preferred method for the results presented in this paper. Following recommendations in literature, the Paule Mandel method is used in conjunction with the Q-Profile approach for obtaining confidence intervals for variance inter-studies [49].

To sum up, how to deal with heterogeneity in a complex scenario like the one posed by preclinical studies is always going to be a complex decision and likely to impact the analysis. The preclinical meta-analysis presented in this article is not expected to be used as a confirmatory test but more as an explorative tool for hypothesis generation. Given the wide availability of methods available nowadays in easy-to-access computational implementations it may be advisable to perform a sensitivity analysis with different methods [52]. The chosen trade-off between the power of the analysis and the rate of false positives may depend on the potential impact of the association identified, and the appetite and capacity of the researchers to act on those hypotheses.

## False discovery rates

One of the purposes of meta-analyses of preclinical data at such scale may be the generation of an ensemble of hypotheses associating targets with events in tissues that can be used as seeds for the generation of new Adverse Outcome Pathways. Ideally, the number of hypotheses generated should be small enough to be actionable and to be pursued for further investigation in a sustainable manner. Hypotheses will be selected by their p-value at a specified level of significance (e.g. $p \leq 0.05$). Of the random effects meta-analyses, 364 produced significant associations, but decisions based on these unadjusted p values are expected to produce a large number of false positives given the large number of meta-analyses carried out. However, any attempt to reduce the number of positives will have the effect of reducing the power of the analysis and increasing the number of false negatives too, potentially missing relevant associations. The Bonferroni correction may be too conservative, especially in scenarios with large number of tests like this (only 38 out of the 364 unadjusted p-values satisfied the Bonferroni adjustment) and, rather than adjusting the results to keep the probability of a false positive below 5%, it may be more useful to adjust the results to keep the proportion of false positives within a set of selected hypotheses below 5% (false discovery rate). One of the most widely used methods to adjust a false discovery rate is the Benjamini-Hochberg method, which produced 121 significant associations.

## Subgroup analyses

The results from the subgroup meta-analyses seem to indicate that in general, there are little differences between subgroups but, in those cases where there are differences, they mainly are at species or at administration route level (S11 Fig). There seems to be little difference between short and long studies, but in the case of duration, only two categories were compared (shorter or equal to 9 days and longer than 9 days). It is also important to keep in mind that the capacity to detect differences between studies with high variability is limited. In that sense, the visual analysis of the forest plot may be more useful to assess the concordance between groups than only a decision based on a p-value.

## Biases

**Data availability.**   As mentioned in the results section, the datasets are very sparse. Not all compounds are measured against all targets and different organisations may have different strategies as to which targets are included in off target pharmacology panels to evaluate potential secondary interactions of their promising compounds, which means more activity data is available for some targets than others. Moreover, results of activity on those targets are likely to influence what will progress to animal studies, which will also bias the composition of data available from animal studies, as compounds with high activities for non-desired off targets will not be as likely to be progressed as compounds that are 'clean'. This also means that most of the compounds for which animal data is available will have low activities for those targets, making it more difficult to find associations. The fact that the whole pharmacology of a compound is rarely known, beyond the limited set of targets in the panels tested, means also that polypharmacology can not properly be evaluated as confounder, and compounds could also be potent against other non-measured targets which could have an impact on the results of the study. The risk of polypharmacology can be minimised in some degree by excluding meta-analyses based on less than a specified number of different compounds (the larger the number of compounds and the more diverse the dataset, the lower the chance the compounds will all hit the same unidentified target).

The strategy used in animal studies is another potential source of bias. It is likely that results in short term studies may affect the progression of an asset to longer term studies, the kind of studies that will be performed or even its termination, affecting the amount and nature of data available.

Additionally, not all the tissues are explored in all studies, with those tissues of higher relevance being analysed in more studies, increasing the data available on them.

**Inclusion criteria and definition of events.**   The inclusion criteria and the definition of the events used in the meta-analyses are another potential source of bias.

In order to simplify the analysis and convert compound activity in a dichotomous variable, the threshold to classify a compound as active is specified in this paper at a pXC50 of 5.5. This value is frequently used in diverse scenarios to classify compounds as binders vs non binders, toxic vs not toxic or active vs not active for classification purposes [53–55]. However, this means that, for example, compounds with activities at micromolar level are considered equally to compounds with nanomolar activity, whereas it is possible that the more active compounds may be more likely to produce events. Additionally, it is also possible that compounds with activities below a pXC50 of 5.5 may produce events if tested at very high concentrations, as it is the case sometimes in preclinical toxicity studies. It should also be considered if different cut-offs need to be applied to different targets of families of targets in case that some require different levels of activation to produce events.

The definition of the event used in this publication attempts to remove subjectivity by considering a lesion as visible or not visible but not considering its severity. This may reduce the power of the analysis, as low severity lesions are more likely to happen in control groups than high severity lesions, but will contribute equally to the calculation of the effect size. Additionally, the meta-analysis is restricted at counting events at tissue level without considering more specific locations of the lesion or its morphology. Although a meta-analysis at higher level of resolution would be desirable, the sample size quickly diminishes when trying to aggregate data at levels more detailed than tissue, given the high number of potential morphologies and locators, making the statistical treatment of the data almost impossible, as the studies are severely underpowered.

Last but not least, the meta-analyses presented do not treat separately the different dosing groups, so it is not possible to assess whether there is a dose effect relationship. Given that these are toxicity studies, doses often are high, and events may be related with nonspecific pharmacology. However, given the different potencies of compounds involved and pharmacokinetic properties, a comparison based on dose would be difficult and it would probably be more interesting to assess whether there is a relation with exposure, with the expectation that events related with a target through a direct mechanism of action will be observed at lower exposures than those due to nonspecific pharmacology. The ability to include toxicokinetic data in a meaningful manner in the meta-analysis would be desirable.

**Study design.**   Fixed effect models will be biased towards the studies with larger sample size as these studies are more likely to produce values with more precision and hence less variance. Studies in rats and longer-term studies tend to have larger sample sizes and so, fixed effect models could potentially bias results towards them.

**Publication bias.**   There is little risk of publication bias, as a systematic approach based on internal data will take in all the studies meeting the inclusion criteria available in the in-house database, which are recorded independently of the outcome of the study. However, there is a risk if an organisation follows different policies to store data for different kinds of studies (e.g. internal studies vs outsourced), or if different ontologies are used in those studies, which may make aggregation of data difficult. It is also possible that not all studies are available in electronic format, introducing another potential source of bias. The introduction of SEND, which

specifies a way to collect and present nonclinical data in a consistent format, is expected to reduce bias in this space. SEND requires a controlled terminology mapping which, for preclinical toxicology, is not as developed or standardised as in the case of human safety (e.g. MedDRA). Several organizations have been working to develop controlled vocabularies and ontologies in this space, such as the International Harmonization of Nomenclature and Diagnostic Criteria for Lesions in Rats/Mice and Non-rodent Species (INHAND) or the ontology developed within the framework of the Innovative Medicines Initiative (IMI) eTOX consortium and also currently used by the eTRANSAFE consortium [56, 57].

## Conclusion

Technological advances in the field of big data have increased our capabilities to query large databases and combine information from different domains and disciplines. With facilitated access to preclinical data and improvements in analytical algorithms there will surely be an expectation for pharmaceutical companies to make sure all the historical data available to them is leveraged to build hypotheses.

Meta-analysis offers a solution to the problem of integrating data across preclinical studies by offering a robust statistical integration and powerful tools to visualise and gather insights from data that are easy to interpret, making it an ideal tool for democratising access to data, and analytics and hypothesis building, in this space. However, it is important to be aware of the potential issues inherent preclinical animal studies (extreme low sample size in some cases), confounding factors and chances of false correlations due to the combining of data that increase the number of combinations. Group analyses to understand the effect of covariates, even if it is just at qualitative level, and control for multiple comparison bias, need to be part of such analyses to reduce the risk of developing hypotheses just by chance. Given the many meta-analysis methods implementations available nowadays and the commoditised access to computer power, it may be advisable to carry out a sensitivity analysis with different methods. In any case, the analyses should be used as exploratory tools, for example, to identify potential target-tissue associations that may lead further investigation to develop causal hypotheses like Adverse Outcome Pathways.

One of the main limitations is, despite the number of preclinical studies available, the sparsity of the data available but this is inherent to the nature of the drug discovery process. In that sense, the aggregative nature of the meta-analyses may also provide a framework for organisations to share preclinical data in the form of tables with counts of events per tissue from studies for compounds known to hit a common target, meta-analyse the results and increase the knowledge about a particular target or develop more robust hypotheses, potentially in precompetitive environment and in the area of toxicology. With that purpose in mind and as example, a file with the counts of the number of events per tissue in the studies involving androgen receptor agonists presented in this publication is available as (S3 File). The R code with the settings used to perform the meta-analysis and to produce the results in Fig 4 and in S8–S10 Figs is also provided (S4 File).

## Supporting information

**S1 File. Initial list of off-targets of interest extracted from literature.**
(TXT)

**S2 File. List of target–tissue associations produced (random effects).**
(XLSX)

**S3 File. Raw data: Number of events per tissue in studies involving androgen agonists.**
(TXT)

**S4 File. R Code with settings used for the meta-analyses.**
(R)

**S1 Fig. Distribution of number of studies by number of animals included in the study.**
(TIF)

**S2 Fig. Distribution of meta-analyses in terms of number of studies included.**
(TIF)

**S3 Fig. Significant target–tissue associations identified per adjustment method: Fixed effect models in blue, random effects models in green.**
(TIF)

**S4 Fig. Plot (log10 scale) comparing observed p values (Y axis) and expected p values (X axis): Line of unity in black.** Line representing unadjusted p value cut-off (0.05) in magenta. Line representing Benjamini-Hockberg 5% false discovery rate (FDR) in green. Line representing Bonferroni adjustment in red. The plot compares the astringencies of the different methods (points above the lines would be considered significant associations after adjustment).
(TIF)

**S5 Fig. Distribution of heterogeneities ($I^2$%) across meta-analyses.**
(TIF)

**S6 Fig. Fixed effect vs random effects odds ratios (ln): Vertical and horizontal represent a value of 2 for the odds ratio (lnOR = 0.693) and line of unity appears as a dashed line.** Numbers in the quadrants separated by the horizontal and vertical lines represent the number of meta-analyses for which the aggregated effect size falls within the quadrant. Datapoints are coloured by the heterogeneity ($I^2$) in the set of studies included in the meta-analysis represented by the datapoint.
(TIF)

**S7 Fig. Androgen agonists subgroup analysis for the effect on number of events in testes (random effects): Forest plots with results by administration route and tests for subgroup differences.**
(TIF)

**S8 Fig. Androgen agonists subgroup analysis for the effect on number of events in testes (random effects): Forest plots with results by duration of the study and tests for subgroup differences.**
(TIF)

**S9 Fig. Androgen agonists-events in testes forest plot (random effects).**
(TIF)

**S10 Fig. Funnel plot (Androgen receptor agonists and events in testes).**
(TIF)

**S11 Fig. Subgroup differences in meta-analysis across covariates (random effects): p-values for the test for differences across groups.** Results are shown for all meta-analysis and also, separately, for meta-analysis with larger effect size (OR > = 2). Results are also disclosed per number of subgroups in on each meta-analysis.
(TIF)

**S1 Checklist.**
(DOC)

## Acknowledgments

The author would like to acknowledge Nicholas Galwey (GSK) for his support throughout the production of this manuscript, Jim Harvey (GSK) for his sponsorship, and Randall Smith (GSK) for the creation of the in-house database of preclinical toxicology studies accessed to obtain data for this research.

## Author Contributions

**Conceptualization:** Jordi Munoz-Muriedas.

**Data curation:** Jordi Munoz-Muriedas.

**Formal analysis:** Jordi Munoz-Muriedas.

**Investigation:** Jordi Munoz-Muriedas.

**Methodology:** Jordi Munoz-Muriedas.

**Project administration:** Jordi Munoz-Muriedas.

**Resources:** Jordi Munoz-Muriedas.

**Software:** Jordi Munoz-Muriedas.

**Supervision:** Jordi Munoz-Muriedas.

**Validation:** Jordi Munoz-Muriedas.

**Visualization:** Jordi Munoz-Muriedas.

**Writing – original draft:** Jordi Munoz-Muriedas.

**Writing – review & editing:** Jordi Munoz-Muriedas.

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
