## [Decision Letter · Decision Letter 0]

31 Mar 2021

PONE-D-20-39813

Large scale meta-analysis of preclinical toxicity data for target characterisation and hypotheses generation

PLOS ONE

Dear Dr. Munoz-Muriedas,

Thank you for submitting your manuscript to PLOS ONE. After careful consideration, we feel that it has merit but does not fully meet PLOS ONE’s publication criteria as it currently stands. Therefore, we invite you to submit a revised version of the manuscript that addresses the points raised during the review process.

We look forward to receiving your revised manuscript.

Kind regards,

Tushar Kanti Dutta, Ph.D.

Academic Editor

PLOS ONE

Additional Editor Comments:

Kindly improve upon the quality of figures as suggested by the reviewer.

Journal Requirements:

Thank you for providing the following Funding Statement: 

"JMM is an employee of GlaxoSmithKline. Apart from my salary, I have received no specific funding for this work."

We note that one or more of the authors is affiliated with the funding organization, indicating the funder may have had some role in the design, data collection, analysis or preparation of your manuscript for publication; in other words, the funder played an indirect role through the participation of the co-authors.

If the funding organization did not play a role in the study design, data collection and analysis, decision to publish, or preparation of the manuscript and only provided financial support in the form of authors' salaries and/or research materials, please review your statements relating to the author contributions, and ensure you have specifically and accurately indicated the role(s) that these authors had in your study in the Author Contributions section of the online submission form. Please make any necessary amendments directly within this section of the online submission form.  Please also update your Funding Statement to include the following statement: “The funder provided support in the form of salaries for authors [insert relevant initials], but did not have any additional role in the study design, data collection and analysis, decision to publish, or preparation of the manuscript. The specific roles of these authors are articulated in the ‘author contributions’ section.”

If the funding organization did have an additional role, please state and explain that role within your Funding Statement.

Please also provide an updated Competing Interests Statement declaring this commercial affiliation along with any other relevant declarations relating to employment, consultancy, patents, products in development, or marketed products, etc. 

We note that you have indicated that data from this study are available upon request. PLOS only allows data to be available upon request if there are legal or ethical restrictions on sharing data publicly. For information on unacceptable data access restrictions, please see http://journals.plos.org/plosone/s/data-availability#loc-unacceptable-data-access-restrictions.

3a) If there are ethical or legal restrictions on sharing a de-identified data set, please explain them in detail (e.g., data contain potentially identifying or sensitive patient information) and who has imposed them (e.g., an ethics committee). Please also provide contact information for a data access committee, ethics committee, or other institutional body to which data requests may be sent.

3b) If there are no restrictions, please upload the minimal anonymized data set necessary to replicate your study findings as either Supporting Information files or to a stable, public repository and provide us with the relevant URLs, DOIs, or accession numbers. Please see http://www.bmj.com/content/340/bmj.c181.long for guidelines on how to de-identify and prepare clinical data for publication. For a list of acceptable repositories, please see http://journals.plos.org/plosone/s/data-availability#loc-recommended-repositories.

Reviewers' comments:

Reviewer's Responses to Questions

**Comments to the Author**

1. Is the manuscript technically sound, and do the data support the conclusions?

Reviewer #1: Yes

2. Has the statistical analysis been performed appropriately and rigorously? 

Reviewer #1: Yes

3. Have the authors made all data underlying the findings in their manuscript fully available?

Reviewer #1: Yes

4. Is the manuscript presented in an intelligible fashion and written in standard English?

Reviewer #1: Yes

5. Review Comments to the Author

Reviewer #1: The figures presented were not upto the mark in terms of resolution. The meta-analysis workflow may be more lucidly explained so as to appeal to more versatile spectrum of readers who wish to apply similar techniques in their own fields.

6. PLOS authors have the option to publish the peer review history of their article (what does this mean?). If published, this will include your full peer review and any attached files.

Reviewer #1: No

---

## [Author Response · Author response to Decision Letter 0]

13 May 2021

Please note all the comments described below and their responses are also included in the "Response to Reviewers" word document enclosed with this submission.

Additional Editor Comments:

Kindly improve upon the quality of figures as suggested by the reviewer.

[JMM] All figures have been remastered from scratch to comply with the resolution and format requirements

Journal Requirements:

 [JMM] Styles have been checked

2. Thank you for providing the following Funding Statement: 

"JMM is an employee of GlaxoSmithKline. Apart from my salary, I have received no specific funding for this work."

We note that one or more of the authors is affiliated with the funding organization, indicating the funder may have had some role in the design, data collection, analysis or preparation of your manuscript for publication; in other words, the funder played an indirect role through the participation of the co-authors.

If the funding organization did not play a role in the study design, data collection and analysis, decision to publish, or preparation of the manuscript and only provided financial support in the form of authors' salaries and/or research materials, please review your statements relating to the author contributions, and ensure you have specifically and accurately indicated the role(s) that these authors had in your study in the Author Contributions section of the online submission form. Please make any necessary amendments directly within this section of the online submission form. Please also update your Funding Statement to include the following statement: “The funder provided support in the form of salaries for authors [insert relevant initials], but did not have any additional role in the study design, data collection and analysis, decision to publish, or preparation of the manuscript. The specific roles of these authors are articulated in the ‘author contributions’ section.”

If the funding organization did have an additional role, please state and explain that role within your Funding Statement.

[JMM] Please find enclosed my updated Funding Statement: Jordi Munoz-Muriedas conceived and designed the analysis, retrieved the data, performed the calculations, prepared conclusions and wrote the manuscript. The funder provided support in the form of salaries for Jordi Munoz-Muriedas but did not have any additional role in the study design, data collection and analysis, decision to publish, or preparation of the manuscript.

Please also provide an updated Competing Interests Statement declaring this commercial affiliation along with any other relevant declarations relating to employment, consultancy, patents, products in development, or marketed products, etc. 

[JMM]Please find enclosed my updated Competing Interests Statement: Jordi Munoz-Muriedas is a full-time employee of GlaxoSmithKline. Jordi Munoz-Muriedas has no other competing interests in relation to other companies, organisations or persons. GlaxoSmithKline is a global healthcare company with a portfolio of medicines in respiratory, HIV, immune-inflammatory and oncology therapeutic areas in addition to vaccines and healthcare products.

[JMM] I can confirm my commercial affiliation does not alter my adherence to all PLOS ONE policies on sharing data and materials. 

 [JMM] I’m the sole author of the paper, only my declared competing interests apply.

3) We note that you have indicated that data from this study are available upon request. PLOS only allows data to be available upon request if there are legal or ethical restrictions on sharing data publicly. For information on unacceptable data access restrictions, please see http://journals.plos.org/plosone/s/data-availability#loc-unacceptable-data-access-restrictions.

[JMM] I believe there may have been a misunderstanding as the submission included raw data and code to allow readers to follow and reproduce the analyses as supporting information and it is my understanding the reviewer found it as per answer to question 3. I can see this misunderstanding may arise from the submission questionnaire where the option “No” implies data is only available upon request. The publication provides raw data and code to allow readers to reproduce the methodology presented. I have updated that option in the submission questionnaire.

The manuscript aims to encourage the use of meta-analyses in preclinical environments, especially in large corporate databases, to leverage all the preclinical data available, but also aims to show how this workflow may be useful in precompetitive environments to combine data from different organisations, and, in that sense, hopefully, encourage more organisations to publicly share preclinical data. It would not be feasible for an organisation to share the whole of its historical proprietary preclinical databases, but for the purposes of this publication, a subset has been shared with 38 animal studies related with Androgen receptor agonists (the example the manuscript goes into more detail). This data allows readers to reproduce all the calculations presented (code with all parameters is also provided as supporting information), and, also very important, it potentially allows them to combine the data with their own preclinical raw data for the same target and enhance the meta-analysis if they wish, or apply the workflow to any other target of their choice. 

3a) If there are ethical or legal restrictions on sharing a de-identified data set, please explain them in detail (e.g., data contain potentially identifying or sensitive patient information) and who has imposed them (e.g., an ethics committee). Please also provide contact information for a data access committee, ethics committee, or other institutional body to which data requests may be sent.

[JMM] The manuscript doesn’t contain any human/clinical data that requires de-identification. All datasets provided are shared with no restrictions.

3b) If there are no restrictions, please upload the minimal anonymized data set necessary to replicate your study findings as either Supporting Information files or to a stable, public repository and provide us with the relevant URLs, DOIs, or accession numbers. Please see http://www.bmj.com/content/340/bmj.c181.long for guidelines on how to de-identify and prepare clinical data for publication. For a list of acceptable repositories, please see http://journals.plos.org/plosone/s/data-availability#loc-recommended-repositories.

[JMM] All relevant data are within the manuscript and its Supporting Information files. A subset of the GSK preclinical database involving the results covered with more detail in the manuscript (results of preclinical studies involving Androgen Receptor agonists) is made available as supporting information. This dataset includes the results of 38 preclinical studies, annotated by species, administration route and duration, with the counts of animals with events observed in control and treated groups by tissue analysed. The data provided allows to reproduce the meta-analyses for the relations involving androgen receptors and provides an idea of the utility of the methodology the paper advocates for. The meta-analyses can be reproduced with the R code also provided.

[JMM] Thanks

[JMM] Bibliography has been checked and to the best of my knowledge is complete and correct. No changes made

Reviewers' comments:

Reviewer's Responses to Questions

Comments to the Author

1. Is the manuscript technically sound, and do the data support the conclusions?

Reviewer #1: Yes

2. Has the statistical analysis been performed appropriately and rigorously?

Reviewer #1: Yes

3. Have the authors made all data underlying the findings in their manuscript fully available?

Reviewer #1: Yes

4. Is the manuscript presented in an intelligible fashion and written in standard English?

Reviewer #1: Yes

5. Review Comments to the Author

Reviewer #1: The figures presented were not upto the mark in terms of resolution. The meta-analysis workflow may be more lucidly explained so as to appeal to more versatile spectrum of readers who wish to apply similar techniques in their own fields.

[JMM] All the figures have been remastered from scratch to increase resolution and comply with the Journal requirements. All figures have been passed through PACE and processed files are provided.

[JMM] The meta-analysis workflow (Fig 1) has been improved using colours to help readers to identify what parts of it where processes, inputs/outputs, and databases. In addition, a few additional components have been added to the workflow to help readers to understand the shape of the data that will be merged and processed during the analysis, and also to indicate points where scientists interact with the process to specify parameters, like for example inclusion criteria or thresholds for activity. I’m particularly grateful for that comment as has certainly helped to improve the figure and will surely have a positive impact.

6. PLOS authors have the option to publish the peer review history of their article (what does this mean?). If published, this will include your full peer review and any attached files.

Do you want your identity to be public for this peer review? For information about this choice, including consent withdrawal, please see our Privacy Policy.

Reviewer #1: No

[JMM] Thanks for the advice. Figures have been processed with PACE. I really found the PACE tool very convenient.

---

## [Editor Report · Decision Letter 1]

18 May 2021

Large scale meta-analysis of preclinical toxicity data for target characterisation and hypotheses generation

PONE-D-20-39813R1

Dear Dr. Munoz-Muriedas,

We’re pleased to inform you that your manuscript has been judged scientifically suitable for publication and will be formally accepted for publication once it meets all outstanding technical requirements.

Kind regards,

Tushar Kanti Dutta, Ph.D.

Academic Editor

PLOS ONE
---

## [Editor Report · Acceptance letter]

20 May 2021

PONE-D-20-39813R1 

Large scale meta-analysis of preclinical toxicity data for target characterisation and hypotheses generation 

Dear Dr. Munoz-Muriedas:

I'm pleased to inform you that your manuscript has been deemed suitable for publication in PLOS ONE. Congratulations! Your manuscript is now with our production department. 

Kind regards, 

on behalf of

Dr. Tushar Kanti Dutta 

Academic Editor

PLOS ONE